# Causal Estimation with Functional Confounders

**Aahlad Puli**[1]
aahlad@nyu.edu

**Adler J. Perotte**[2]
adler.perotte@columbia.edu

**Rajesh Ranganath**[1,3]
rajeshr@cims.nyu.edu

[1]Computer Science, New York University, New York, NY 10011
[2]Biomedical Informatics, Columbia University, New York, NY 10032
[3]Center for Data Science, New York University, New York, NY 10011

## Abstract

Causal inference relies on two fundamental assumptions: *ignorability* and *positivity*. We study causal inference when the true confounder value can be expressed as a function of the observed data; we call this setting *estimation with functional confounders (*EFC*)*. In this setting ignorability is satisfied, however positivity is violated, and causal inference is impossible in general. We consider two scenarios where causal effects are estimable. First, we discuss interventions on a part of the treatment called *functional interventions* and a sufficient condition for effect estimation of these interventions called *functional positivity*. Second, we develop conditions for nonparametric effect estimation based on the gradient fields of the functional confounder and the true outcome function. To estimate effects under these conditions, we develop Level-set Orthogonal Descent Estimation (LODE). Further, we prove error bounds on LODE's effect estimates, evaluate our methods on simulated and real data, and empirically demonstrate the value of EFC.

## 1 Introduction

Determining the effect of interventions on outcomes using observational data lies at the core of many fields like medicine, economic policy, and genomics. For example, policy makers estimate effects to elect whether to invest in education or job training programs. In medicine, doctors use effects to design optimal treatment strategies for patients. Geneticists perform genome-wide association studies (GWAS) to relate genotypes and phenotypes. In observational data, there could exist unobserved variables that affect both the intervention and the outcome, called confounders. A necessary condition for the causal effect to be identified is that all confounders are observed; called *ignorability*. If ignorability holds, a sufficient condition for causal effect estimation is adequate variation in the intervention after conditioning on the confounders; this condition is called *positivity*.

The data apriori does not differentiate between confounders and interventions. It is the practitioners that select interventions of interest from all pre-outcome variables (variables that occur before the outcome). Then, assuming knowledge of the data generating mechanism, practitioners can label certain variables amongst the remaining pre-outcome variables as confounders. This corresponds to indexing into the set of pre-outcome variables.

In certain problems the confounders are specified as a function of the pre-outcome variables that does not simply index into the set of pre-outcome variables. For a concrete example, consider GWAS. The goal in GWAS is to estimate the influence of genetic variations on phenotypes like disease risk. In GWAS, population and family structures both result in certain genetic variations and affect phenotypes and therefore, are confounders [4]. Practitioners specify these confounders by using the genetic similarity between individuals [15, 19, 31], which is a function of the genetic variations. When the confounders are a function of the same pre-outcome variables that define the interventions, positivity is violated. Then, the class of interventions whose effects are estimable is not well-defined.

We study causal effect estimation in such settings, where a function of the pre-outcome variables provides the confounder and these same pre-outcome variables define the intervention. We call this estimation with functional confounders (EFC). In EFC, one column in the observed data is the outcome and all others are pre-outcome variables. We assume access to a function $h(\cdot)$ that takes as input the pre-outcome variables and returns the value of the confounder. Further, we assume these confounders give us ignorability. In settings like GWAS, the function $h$ reflects the practitioner-specified function that captures the genetic variation influenced by the population structure. In traditional observational causal inference (OBS-CI), $h(\cdot)$ reflects the selection of certain variables in the data and labelling them as confounders. In EFC, two different values of the confounder are never observed for the same setting of the pre-outcome variables. This means that positivity is violated and the effects of only certain interventions may be estimable.

We address this issue in two ways. First, we investigate a class of plausible interventions that are *functions* of the observed pre-outcome variables, called functional interventions. We develop a sufficient condition to estimate the effects of said functional interventions, called functional positivity (F-POSITIVITY). Second, we consider intervening on all pre-outcome variables, called the *full* intervention. We develop a sufficient condition to estimate the effect of the *full* intervention, called causal redundancy (C-REDUNDANCY). For an intervention, given a confounder value, C-REDUNDANCY allows us to compute a surrogate intervention such that the conditional effect of the surrogate is equal to that of the original intervention. We also show that such surrogate interventions exist only under a certain condition that we call Effect Connectivity, that is necessary for nonparametric effect estimation in EFC. This condition is satisfied by default in traditional OBS-CI if ignorability and positivity hold. Then, we develop an algorithm for causal estimation assuming C-REDUNDANCY, called Level-set Orthogonal Descent Estimation (LODE), which estimates effects using surrogate interventions. If the surrogate is not estimated well, LODE's estimates are biased. We establish bounds on this bias that capture the mitigating effect of the smoothness of the true outcome function.

**Related work**   The problem of genome-wide association studies (GWAS) is to estimate the effect of genetic variations(also called single nucleotide polymorphisms (SNPs)) on the phenotype [29]. The ancestry of the subjects acts as a confounder in GWAS. In GWAS practice, principle component analysis (PCA) and linear mixed models (LMMs) are used to compute this confounding structure [19, 31]. Lippert et al. [15] suggest estimating the confounders and effects on *separate* subsets of the SNPs. This separation disregards the confounding that is captured in the interaction of the two subsets of SNPs. GWAS is a special case of effects from multiple treatments (MTE) where the confounder value is specified via optimization as a function of the pre-outcome variables [20, 30]. In all these settings, positivity is violated and not all effects are estimable. We provide an avenue for nonparametric effect-estimation of the full intervention under a new condition, C-REDUNDANCY.

**Traditional observational causal inference (OBS-CI) review**   We setup causal inference with Structural Causal Models [17] and use $do(t = t^*)$ to denote making an intervention. Let $t$ be a vector of the interventions, $z$ be the confounder, and $y$ be the outcome. Let $\eta \sim p(\eta)(\eta \perp\!\!\!\perp (z, t))$ be noise. With $f$ as the *outcome function*, we define the causal model for traditional OBS-CI as [1]:

$$z \sim p(z), \quad t \sim p(t \mid z), \quad y = f(t, z, \eta).$$

Let $p(y, z, t)$ denote the joint distribution implied by this data generating process. The effects of interest under the full intervention $do(t = t^*)$ are the average and *conditional effect*

$$(\text{average}) \quad \tau(t^*) = \mathbb{E}_{z,\eta} f(t^*, z, \eta) \qquad (\text{conditional}) \quad \phi(t^*, z) = \mathbb{E}_\eta \left[ f(t^*, z, \eta) \right]. \qquad (1)$$

With observed confounders, two assumptions make causal estimation possible: *ignorability* and *positivity*. Ignorability means that *all* confounders $z$ are observed in data. Conditioning on all the confounders, the outcome under an intervention is distributed as if conditional on the value of the intervention: $p(y = y_1 \mid do(t = t^*), z = z) = p(f(t^*, z, \eta) = y_1) = p(y = y_1 \mid t = t^*, z = z)$. This allows the expression of average effect as an expectation over the *observed* outcomes $\tau(t^*) = \mathbb{E}_{z,\eta}[f(t^*, z, \eta)] = \mathbb{E}_z \mathbb{E}[y \mid z, t^*]$. The conditional expectation only exists for all $t^*$ if $p(y \mid z, t = t^*) = {}^{p(y,z,t=t^*)}\!/{}_{p(z)p(t=t^* \mid z)}$ exists. *Positivity* guarantees this existence

$$(\text{positivity}) \quad \forall t^* \in \text{supp}(t) \quad p(z = z) > 0 \implies p(t = t^* \mid z = z) > 0. \qquad (2)$$

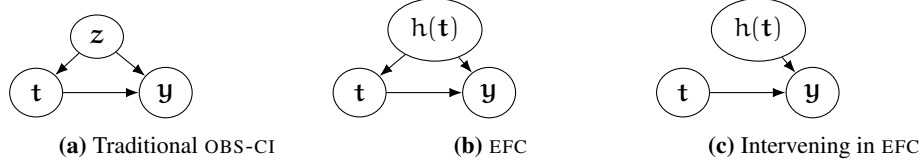

**(a)** Traditional OBS-CI      **(b)** EFC      **(c)** Intervening in EFC

**Figure 1:** Causal Graphs for Traditional OBS-CI vs. EFC.

## 2   Estimation with functional confounders

In traditional OBS-CI, causal estimation relied on knowing the confounders. In this section, we consider settings where confounders are known via a function of the pre-outcome variables $h(\mathbf{t}) = \mathbf{z}$. We call this setting *estimation with functional confounders (*EFC*)*. An example of this is GWAS, where SNPs (the pre-outcome variables) are used to estimate the confounding population structure through methods like PCA [31]. Assuming the confounders are a function of the pre-outcome variables violates positivity in general. Positivity is violated in this setting because

$$\forall t_1, t_2 \in \text{supp}(\mathbf{t}) \ \text{ s.t. } \ h(t_2) \neq h(t_1) \implies p(\mathbf{z} = h(t_2) \mid \mathbf{t} = t_1) = 0 \neq p(\mathbf{z} = h(t_2)) > 0$$

In words, two different confounder values cannot occur for the same $\mathbf{t}$. A positivity violation precludes nonparametric effect estimation of the full intervention $do(\mathbf{t} = \mathbf{t}^*)$.

**Positivity and Regression Identifiability**   Positivity can be viewed as providing identifiability. To see this, let the confounder be $\mathbf{z} = h(\mathbf{t})$ and the outcome be $\mathbf{y}(\mathbf{t}, \mathbf{z}, \boldsymbol{\eta}) = \mathbf{z} + h(\mathbf{t})$. Now consider regressing $\mathbf{z}$ and $\mathbf{t}$ onto $\mathbf{y}$. Then, functions $\mathbf{y} = \alpha \mathbf{z} + \beta h(\mathbf{t})$ indexed by $\alpha, \beta$, such that $\alpha + \beta = 2$, are consistent with the observed data. Thus, there exist infinitely many solutions to the conditional expectation of $\mathbf{y}$ on $(\mathbf{t}, \mathbf{z})$, meaning that the regression is not identifiable. Assuming positivity necessitates sufficient randomness to identify the regression and thus the causal effect. A violation of positivity means that nonparametric estimation of causal effects needs further assumptions.

### 2.1   Setup for EFC

In EFC, the confounder is provided as a non-bijective function $h$ of the pre-outcome variables $\mathbf{t}$. To reflect this property, we use $h(\mathbf{t})$ to denote the confounder. As an illustrative example, let $\mathcal{G}$ be the Gamma distribution and consider $\mathbf{z} \in \{-1, 1\}, p(\mathbf{z} = 1) = 0.5$ is the confounder and the intervention of interest is $\mathbf{t} = \mathbf{z} * \mathcal{G}(1, \exp(\mathbf{z}))$. Note $\text{sign}(\mathbf{t}) = \mathbf{z}$ meaning that $h(\mathbf{t}) = \text{sign}(\mathbf{t})$ is the confounder. Figure 1 shows causal graphs connecting our EFC notation to that in traditional OBS-CI. With noise $\boldsymbol{\eta} \sim p(\boldsymbol{\eta})(\boldsymbol{\eta} \perp\!\!\!\perp \mathbf{t})$, our causal model samples, in order, the confounder "part" of pre-outcome variables $h(\mathbf{t})$, the pre-outcome variables $\mathbf{t}$, and the outcome $\mathbf{y}$ via the *outcome function* $f$ [2]:

$$h(\mathbf{t}) \sim p(h(\mathbf{t})) \quad \mathbf{t} \sim p(\mathbf{t} \mid h(\mathbf{t})) \quad \mathbf{y} = f(\mathbf{t}, h(\mathbf{t}), \boldsymbol{\eta})$$

Similar to traditional OBS-CI, for an intervention $\mathbf{t}^*$ the average effect, $\tau(\cdot)$, and the conditional effect, $\phi(\cdot, \cdot)$ at $h(t_2^*)$, respectively, are defined as:

$$\tau(\mathbf{t}^*) = \mathop{\mathbb{E}}_{h(\mathbf{t}), \boldsymbol{\eta}} [f(\mathbf{t}^*, h(\mathbf{t}), \boldsymbol{\eta})], \qquad \phi(\mathbf{t}^*, h(t_2^*)) = \mathop{\mathbb{E}}_{\boldsymbol{\eta}}[f(\mathbf{t}^*, h(t_2^*), \boldsymbol{\eta})]. \qquad (3)$$

As the pre-outcome variables determine the confounder, positivity is violated. Further, the *outcome function* $f(\mathbf{t}, h(\mathbf{t}), \boldsymbol{\eta})$ could recover the exact value of $h(\mathbf{t})$ from $\mathbf{t}$ instead of its second argument. Thus, two different outcome functions could lead to the same observational data distribution, posing a fundamental obstacle to causal effect estimation. This is the central challenge in EFC.

### 2.2   Causal Questions With Functional Positivity

Without positivity, we can only estimate the effects of certain functions of $\mathbf{t}$. We call such interventions, on some function $g(\mathbf{t})$, *functional interventions*. The implied causal model for the outcome for functional intervention value $g(\mathbf{t}^*)$ and confounder value $h(t_2^*)$ is first $\mathbf{t} \sim p(\mathbf{t} \mid g(\mathbf{t}) = g(\mathbf{t}^*), h(\mathbf{t}) = h(t_2^*))$ and then $\mathbf{y} = f(\mathbf{t}, h(t_2^*), \boldsymbol{\eta})$ [3]. Then, the *functional* average effect is

$$(\text{average}) \quad \tau(g(\mathbf{t}^*)) = \mathbb{E}_{h(\mathbf{t}), \boldsymbol{\eta}} \mathbb{E}_{\mathbf{t} \mid g(\mathbf{t}) = g(\mathbf{t}^*), h(\mathbf{t})} [f(\mathbf{t}, h(\mathbf{t}), \boldsymbol{\eta})].$$

An example of a functional intervention is intervening on the cumulative dosage of a drug. In contrast, traditional interventions would set each individual dose given at different points in time.

**F-POSITIVITY and Functional Effect Estimation** For the causal model above to be well-defined for all functional interventions $g(t^*)$, the conditional $p(t \mid g(t) = g(t^*), h(t) = h(t_2^*))$ must exist. To guarantee this existence, we define *functional positivity (F-POSITIVITY)* for any $g(t^*)$

$$(\text{F-POSITIVITY}) \quad p(h(t) = h(t_2^*)) > 0 \implies p(g(t) = g(t^*) \mid h(t) = h(t_2^*)) > 0. \tag{4}$$

F-POSITIVITY says that the function of the pre-outcome variables that is being intervened on needs to have sufficient randomness when the function of the pre-outcome variables that defines the confounders is fixed. Further, under F-POSITIVITY, effect estimation for functional interventions is reduced to traditional OBS-CI on data $p(y, g(t), h(t))$. With positivity and ignorability satisfied, traditional causal estimators such as propensity scores [23], matching [21], regression [11], and doubly robust methods [22] can be used to estimate the causal effect. Focusing on regression, let $f_\theta$ be a flexible function, then $\min_\theta \mathbb{E}_{y,t}[(y - f_\theta(h(t), g(t)))^2]$ would estimate the conditional expectation of interest : $\mathbb{E}[y \mid h(t), g(t^*)]$. With $\theta$, the effect of $g(t^*)$ can be estimated by averaging the estimate of the conditional expectation over the marginal distribution $p(h(t))$:

$$\tau(g(t^*)) = \mathbb{E}_t[f_\theta(h(t), g(t^*))]. \tag{5}$$

## 3 Identification of effects of the full intervention

When positivity is violated, causal effects cannot be estimated as conditional expectations over the observed data in general. We give a functional condition, called causal redundancy (C-REDUNDANCY), that allows us to estimate the effect of the full intervention $do(t = t^*)$, even when positivity is violated. Specifically, C-REDUNDANCY allows us to construct a *surrogate intervention* $t'(t^*, h(t_2^*))$ whose conditional effect at $h(t')$ matches the conditional effect of interest, $\phi(t^*, h(t_2^*))$. Let $\tilde{t}$ be a fixed value of the full intervention, then C-REDUNDANCY is

**Assumption.** *Recall the outcome* $y = f(\tilde{t}, h(\tilde{t}), \eta)$. *With* $\nabla_{\tilde{t}}$ *as gradient w.r.t. to argument* $\tilde{t}$:

$$\forall \tilde{t}, h(\tilde{t}_2), \eta, \quad \nabla_{\tilde{t}} f(\tilde{t}, h(\tilde{t}_2), \eta)^\top \nabla_{\tilde{t}} h(\tilde{t}) = 0.$$

In words, C-REDUNDANCY is the condition that the outcome function $f$ uses the value of the confounder from its second argument instead of computing $h(t)$ from the first argument[4]. To compute the conditional effect $\phi(t^*, h(t_2^*))$, we develop Level-set Orthogonal Descent Estimation (LODE). LODE's key step is to construct a surrogate intervention $t'(t^*, h(t_2^*))$ such that

$$\phi(t^*, h(t_2^*)) = \phi(t'(t^*, h(t_2^*)), h(t_2^*)), \qquad h(t_2^*) = h(t'(t^*, h(t_2^*))).$$

By definition, a surrogate intervention lives in the conditional effect level-set: $\{\tilde{t} : \phi(\tilde{t}, h(t_2^*)) = \phi(t^*, h(t_2^*))\}$. So LODE searches this level-set for $t'(t^*, h(t_2^*))$. See fig. 2 which plots the conditional effect level-sets with the value of $h(t)$ fixed (red) in $(\text{supp}(t), \text{supp}(h(t)))$-space. Green corresponds to the observed data, $\text{supp}(t, h(t))$. LODE finds $t'(t^*, h(t_2^*))$ by traversing the level-sets (**black**) to account for the confounder part mismatch $h(t^*) \neq h(t_2^*)$. C-REDUNDANCY ensures LODE can traverse these level-sets as it implies $\nabla_{\tilde{t}} \phi(\tilde{t}, h(\tilde{t}_2)) \nabla_{\tilde{t}} h(\tilde{t}) = 0$ under the

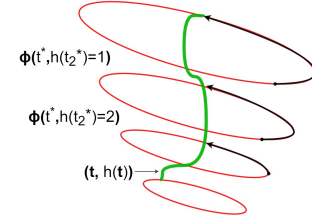

$\phi(t^*, h(t_2^*)=1)$

$\phi(t^*, h(t_2^*)=2)$

$(t, h(t))$

**Figure 2:** LODE's traversal.

regularity conditions in theorem 1. Thus, under C-REDUNDANCY, surrogate interventions can be constructed by solving a gradient flow equation which guarantees identification as follows:

**Theorem 1.** *Assume* C-REDUNDANCY *holds. Assuming the following:*

1. *Let* $t'(t^*, h(t_2^*))$ *be the limiting solution to the gradient flow equation* $\frac{d\tilde{t}(s)}{ds} = -\nabla_{\tilde{t}}(h(\tilde{t}(s)) - h(t_2^*))^2$, *initialized at* $\tilde{t}(0) = t^*$; *i.e.* $t'(t^*, h(t_2^*)) = \lim_{s \to \infty} \tilde{t}(s)$. *Further, let* $h(t'(t^*, h(t_2^*))) = h(t_2^*)$ *and* $t'(t^*, h(t_2^*)) \in supp(t)$.

2. $f(\tilde{t}, h(\tilde{t}), \eta)$ *and* $h(\tilde{t})$ *as functions of* $\tilde{t}, h(\tilde{t})$ *are continuous and differentiable and the derivatives exist for all* $\tilde{t}, \eta$. *Let* $\nabla_{\tilde{t}} f(\tilde{t}, h(\tilde{t}), \eta)$ *exist and be bounded and integrable w.r.t. the probability measure corresponding to* $p(\eta)$, *for all values of* $\tilde{t}$ *and* $h(\tilde{t})$.

*Then the conditional effect (and therefore the average effect) is identified:*

$$\phi(t^*, h(t_2^*)) = \phi\left(t'(t^*, h(t_2^*)), h(t'(t^*, h(t_2^*)))\right) = \mathbb{E}\left[y \mid t = t'(t^*, h(t_2^*))\right] \quad (6)$$

In words, the key idea is that starting at $\tilde{t}(0) = t^*$ and following $\nabla_{\tilde{t}} h(\tilde{t})$ means $\tilde{t}(s)$ always lies in the level-set $\{\tilde{t} : \phi(\tilde{t}, h(t_2^*)) = \phi(t^*, h(t_2^*))\}$. See appendix A.2 for the proof. While C-REDUNDANCY is stated in terms of the gradient of the outcome function, it suffices for theorem 1 to assume a weaker condition about the gradient of the conditional effect: $\nabla_{\tilde{t}} \mathbb{E}_{\eta} f(\tilde{t}, \tilde{t}_2, \eta)^\top \nabla_{\tilde{t}} h(\tilde{t}) = 0$.

**Surrogate Positivity** In theorem 1, we assumed that the surrogate $t'(t^*, h(t_2^*)) \in \text{supp}(t)$. This condition, which we call surrogate positivity (analogous to positivity), states that for any intervention and confounder, surrogate interventions that are limiting solutions to the gradient flow equation have nonzero density conditional on the confounder value. Formally, for any intervention $t = t^*$

$$p(h(t) = h(t_2^*)) > 0 \implies p(t = t'(t^*, h(t_2^*)) \mid h(t) = h(t_2^*)) > 0, \quad (7)$$

and $t'(t^*, h(t_2^*))$ satisfies assumption 1 in theorem 1. Surrogate positivity along with C-REDUNDANCY, is sufficient for full effect estimation under EFC. Next, we show that the positivity assumption in traditional causal inference is a special case of surrogate positivity.

**Traditional observational causal inference (OBS-CI) and LODE** Let the confounder and intervention of interest in traditional OBS-CI be $z$ and $a$ respectively. Assume both are scalars and ignorability and positivity hold. This setup can be embedded in EFC by defining the vector of pre-outcome variables as: $t = [a; z]$. In this setting, C-REDUNDANCY and surrogate positivity(eq. (7)) hold by default. Let the outcome be $y = f(t, h(t)) = f(a, z)$, where $f$ only depends on the first element of $t$, i.e. $a$[5]. Let $e_1 = [1, 0]$ and $e_2 = [0, 1]$. In traditional OBS-CI as EFC, $\nabla_{\tilde{t}} f(\tilde{t}, h(t_2^*)) \propto e_1$ and $\nabla_{\tilde{t}} h(\tilde{t}) \propto e_2$ meaning that $\nabla_{\tilde{t}} f(\tilde{t}, h(t_2^*))^\top \nabla_{\tilde{t}} h(\tilde{t}) = 0$. Thus, C-REDUNDANCY holds by default. Moreover, under positivity of $a$ w.r.t. $z$, we also have surrogate positivity for traditional OBS-CI as an EFC problem. In this setting, LODE computes $t' = [a^*, h(t_2^*)]$ by following $-\nabla_{\tilde{t}} h(\tilde{t}) = [0, -1]$, which only changes the value of $h(\tilde{t}_2)$, not the value of $a$. Thus, $t^*$ and $t'(t^*, h(t_2^*))$ will have the same first element and $t'$'s second element will be $h(t_2^*)$. As $a$ has positivity w.r.t. $z$, we have $p(a = a^*, z = h(t_2^*)) > 0$ which means $t' \in \text{supp}(\tilde{t})$. The estimated conditional effect is $\mathbb{E}[y \mid t = t'(t^*, h(t_2^*))] = f([a^*, z^*], h(t_2^*)) = \mathbb{E}[y \mid a = a^*, z = h(t_2^*)]$, which matches the estimate in traditional OBS-CI.

**Implementation of LODE** LODE first estimates the conditional expectation $\mathbb{E}[y \mid t]$; this can be done with model-based or nonparametric estimators. This is achieved by regressing $y$ on $t$, $\hat{f} = \arg\min_{u \in \mathcal{F}} \mathbb{E}_{y, t \sim D}(y - u(t))^2$, with empirical distribution $D$. The surrogate intervention $t'(t^*, h(t_2^*))$ is computed using Euler integration to solve the gradient flow equation. Euler integration in this setting is equivalent to gradient descent with a fixed step size. Other, more efficient schemes like Runge–Kutta numerical integration methods [3] could also be used. The conditional effect estimate is $\hat{f}(t'(t^*, h(t_2^*)))$. See algorithm 1 for a description.

### 3.1 Estimation error of LODE in practice

To compute the surrogate intervention $t'$, LODE uses the gradients of $h(\cdot)$ in Euler integration. In practice, taking Euler integration steps, instead of solving the gradient flow exactly, could result in errors. Then $t'$ could lie outside the level-set of the conditional effect $\phi(t^*, h(t_2^*)) = \mathbb{E}_{\eta}[f(t^*, h(t_2^*), \eta)]$. Further, if $h(t'(t^*, h(t_2^*))) \neq h(t_2^*)$, LODE incurs error for conditioning on a value of the confounder that is different from $h(t_2^*)$. The error due to $t'$ estimation is decoupled from the error in the estimation of $\mathbb{E}[y \mid t]$ which adds without further amplification. We formalize this error:

**Theorem 2.** *Consider the conditional effect* $\phi(t^*, h(t_2^*))$. *Let* $\hat{t}(t^*, h(t_2^*))$ *be the estimate of the surrogate intervention computed by* LODE*, computed via Euler integration of the gradient flow* $\frac{d\tilde{t}(s)}{ds} = -\nabla_{\tilde{t}}(h(\tilde{t}(s)) - h(t_2^*))^2$, *initialized at* $\tilde{t}(0) = t^*$. *Assume the true surrogate* $t'(t^*, h(t_2^*))$ *exists and is the limiting solution to the gradient flow equation.*

1. *Let the finite sample estimator of* $\mathbb{E}[y \mid t = \tilde{t}]$ *be* $\hat{f}(\tilde{t})$. *Let the error for all* $\tilde{t}$ *be bounded,* $|\hat{f}(\tilde{t}) - \mathbb{E}[y \mid t = \tilde{t}]| \leqslant c(N)$, *where* $N$ *is the sample size and* $\lim_{N \to \infty} c(N) = 0$.

2. *Assume* $K$ *Euler integrator steps were taken to find the surrogate estimate* $\hat{t}(t^*, h(t_2^*))$, *each of size* $\ell$. *Let the maximum confounder mismatch be* $\max_{i \leqslant K}(h(\tilde{t}_i) - h(t_2^*))^2 = M$.

3. *Let $L_{z,\tilde{t}}$ be the Lipschitz-constant of $\phi(\tilde{t}, h(\tilde{t}_2))$ as a function of $h(\tilde{t}_2)$, for fixed $\tilde{t}$.*
   *Let $L_e$ be the Lipschitz-constant of $\mathbb{E}[y \mid t = \tilde{t}] = \phi(\tilde{t}, h(\tilde{t}))$ as a function of $\tilde{t}$.*
   *Assume $h$ has a gradient with bounded norm, $\|\nabla h(\tilde{t})\|_2 < L_h$.*
   *Assume $f$'s Hessian has bounded eigenvalues: $\forall \tilde{t}, \tilde{t}_2, \ \|\nabla_{\tilde{t}}^2 \phi(\tilde{t}, h(\tilde{t}_2))\|_2 \leqslant \sigma_{H\phi}$.*

*The conditional effect estimate error, $\xi(t^*, h(t_2^*)) = |\hat{f}(\hat{t}) - \phi(t^*, h(t_2^*))|$, is upper bounded by:*

$$c(N) + \min\left(L_e \|t' - \hat{t}\|_2, \ 2K\ell^2\left(\mathcal{O}(\ell) + M\sigma_{H\phi}L_h^2\right) + L_{z,\hat{t}}\|h(\hat{t}) - h(t_2^*)\|_2\right) \tag{8}$$

See appendix A.3 for the proof. Theorem 2 captures the trade-off between biases due to conditioning on the wrong confounder value and due to the accumulated error in solving the gradient flow equation. This accumulated error analysis may be loose in settings where the sum of many gradient steps lead to $\hat{t} \approx t'$, even if each step individually induces large error. In such settings, the term that depends on $\|\hat{t} - t'\|_2$ is a better measure of error. The maximum-mismatch $M$ appears because Euler integrator takes steps that depend on the magnitude of the gradient which depends on the mismatch value $(h(\tilde{t}_i) - h(t_2^*))$. If mismatch is large for some $i$, the Euler step could lead to a large error for a fixed step size $\ell$. We discuss the assumptions in theorems 1 and 2 in appendix A.1

### 3.2 Effect Connectivity and the Existence of $t'(t^*, h(t_2^*))$

The key element in Theorem 1 is the surrogate intervention $t'$ such that its conditional effect given $h(t')$, equals that of $t^*$ and $h(t_2^*)$. The orthogonality $\nabla_{\tilde{t}} f^\top \nabla_{\tilde{t}} h = 0$, is a functional condition that does not guarantee $t'(t^*, h(t_2^*))$ exists in $\text{supp}(t)$; a necessity to compute $\mathbb{E}[y \mid t = t']$ without additional parametric assumptions. We give a general condition called *Effect Connectivity* that guarantees the surrogate intervention exists. With conditional effect $\phi(t^*, h(t_2^*))$, for any $t^*$

$$p(h(t) = h(t_2^*)) > 0 \implies p(\phi(t, h(t)) = \phi(t^*, h(t_2^*)) \mid h(t) = h(t_2^*)) > 0. \tag{9}$$

In words, $t$ has a chance of setting the conditional effect to any possible value $\text{supp}(\phi(t, h(t_2)))$ given any confounder value $h(t_2^*) \in \text{supp}(h(t))$. An equivalent statement is that every level set of the conditional effect $\phi(t^*, h(t_2^*))$, with $h(t_2^*)$ fixed, contains an intervention for each confounder value. That is, for some $h(t_2^*)$ define the level set $A_c = \{t^*; f(t^*, h(t_2^*)) = c\}$, then $\forall h(t_2^*) \in \text{supp}(h(t)), \ p(t \in A_c \mid h(t) = h(t_2^*)) > 0$.

**Theorem 3.** *Under Effect Connectivity, eq. (9), any surrogate intervention $t'(t^*, h(t_2^*)) \in \text{supp}(t)$.*

We give the proof in appendix A.4. Whether the intervention $t'(t^*, h(t_2^*))$ can be found via tractable search is problem-specific. If the surrogate $t'(t^*, h(t_2^*))$ exists $\forall t^*, h(t_2^*)$, then eq. (9) holds by definition of the surrogate. Effect Connectivity allows us to reason about values of $f$ anywhere in $\text{supp}(t) \times \text{supp}(h(t))$ using only samples from $p(y, t)$. Further, it is necessary in EFC:

**Theorem 4.** *Effect Connectivity is necessary for nonparametric effect estimation in EFC.*

We prove this in appendix A.5. Effect Connectivity ensures that causal models with different causal effects have different observational distributions. Then, parametric assumptions on the causal model are not necessary to estimate effects.

## 4 Experiments

We evaluate LODE on simulated data first and show that LODE can correct for confounding. We also investigate the error induced by imperfect estimation of the surrogate intervention in LODE. Further, we run LODE on a GWAS dataset [6] and demonstrate that LODE is able to correct for confounding and recovers genetic variations that have been reported relevant to Celiac disease [8, 25, 14, 1].

### 4.1 Simulated experiments

We investigate different properties of LODE on simulated data where ground truth is available. Let the dimension of $t$ (pre-outcome variables) be $T = 20$ and outcome noise be $\eta \sim \mathcal{N}(0, 0.1)$. We consider two EFC causal models, denoted by A and B with different $h(t)$ and $f(t, h(t), \eta)$:

$$(A) \quad h(t) = \gamma \frac{\sum_i t_i}{\sqrt{T}}, \qquad t \sim \mathcal{N}(0, \sigma^2 \mathbb{I}^{T \times T}), \quad y = \frac{\sum_i (-1)^i t_i}{\sqrt{T}} + \alpha h(t)^2 + (1 + \alpha)h(t) + \eta$$

$$(B) \quad h(t) = \sum_{i:i \in 2\mathbb{Z}} \gamma t_i t_{i+1}, \quad t \sim \mathcal{N}(0, \sigma^2 \mathbb{I}^{T \times T}), \quad y = \frac{\sum_i (-1)^i t_i^2}{\sqrt{T}} + \alpha h(t) + \eta$$

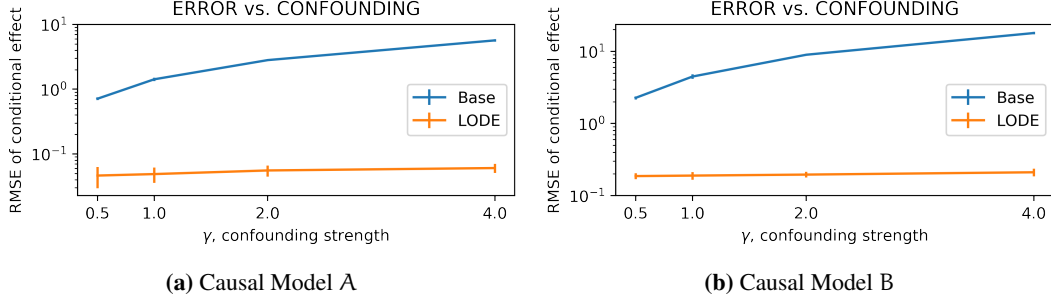

**(a)** Causal Model A                      **(b)** Causal Model B

**Figure 3:** RMSE of estimated conditional effect vs. strength of confounding $\gamma$. LODE corrects for confounding and produces good effect estimates across different values of $\gamma$.

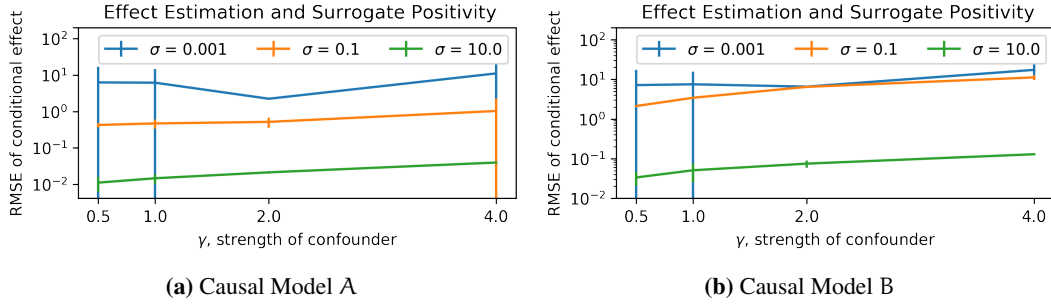

**(a)** Causal Model A                      **(b)** Causal Model B

**Figure 4:** RMSE of estimated conditional effect estimate vs. the strength of confounding $\gamma$, for different levels of variance of $\mathbf{t}$, $\sigma^2$. Small $\sigma$ leads to large conditional estimation error.

In both causal models, C-REDUNDANCY is satisfied. The constant $\gamma$ controls the strength of the confounder and the constant $\alpha$ controls the Lipschitz constant of the outcome as a function of the confounder. We let the variance $\sigma^2 = 1$, unless specified otherwise. In the following, we train on 1000 samples and report conditional effect root-mean-squared error (RMSE), computed with another 1000 samples. We used a degree-2 kernel ridge regression to fit the outcome model as a function of $\mathbf{t}$. This model is correctly specified, and so the conditional $\mathbb{E}[\mathbf{y} \mid \mathbf{t} = \tilde{\mathbf{t}}]$ can be estimated well. We compare against a baseline estimate of conditional effect that is the same outcome model's estimate of $\mathbb{E}[\mathbf{y} \mid \mathbf{t} = \mathbf{t}^*]$. This baseline fails to account for confounding and produces a biased estimate of the conditional effect of $\mathrm{do}(\mathbf{t} = \mathbf{t}^*)$, conditional on any $h(\mathbf{t}_2^*) \neq h(\mathbf{t}^*)$.

First, we investigate how well LODE can correct for confounding for both causal models. We let $\alpha = 1$ and obtain surrogate estimates by Euler integrating until the quantity $\mathbb{E}_{\mathbf{t}^*, h(\mathbf{t}_2^*)}(h(\tilde{\mathbf{t}}(s)) - h(\mathbf{t}_2^*))^2$ is smaller than $10^{-4}$ times value at initialization, where $\mathbb{E}_{\mathbf{t}^*, h(\mathbf{t}_2^*)}$ is expectation over the evaluation set. In fig. 3, we plot the mean and standard deviation of conditional effect RMSE averaged over 10 seeds, for different strengths of confounding. We see that LODE is able to estimate effects well across multiple strengths of confounding while the baseline suffers.

Second, we investigate LODE's estimation when surrogate positivity holds but the probability $p(\mathbf{t} \approx \mathbf{t}'(\mathbf{t}^*, h(\mathbf{t}_2^*)))$ is very small. This results in estimation error due to poor fitting of the outcome model in low density regions of $\mathrm{supp}(\mathbf{t})$. We run LODE on simulated data where $\mathbf{t}$ is generated with different variances ($\sigma^2$). For small $\sigma$, the outcome model error is large when using surrogate interventions $\mathbf{t}'(\mathbf{t}^*, h(\mathbf{t}_2^*))$, where either $h(\mathbf{t}_2^*)$ or $\mathbf{t}^*$ is large. This leads to high variance effect estimation as we show in fig. 4 for both causal models. For various variances of $\mathbf{t}$, $\sigma^2$, we plot the mean and standard deviation of RMSE of estimated conditional effect over 10 seeds, against different $\gamma$.

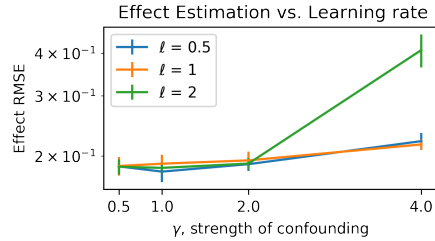

**Figure 5:** RMSE of estimated conditional effect vs. step size in Euler Integrator in causal model B. Accumulating error due to large step size in Euler integrator increases with strength of confounding.

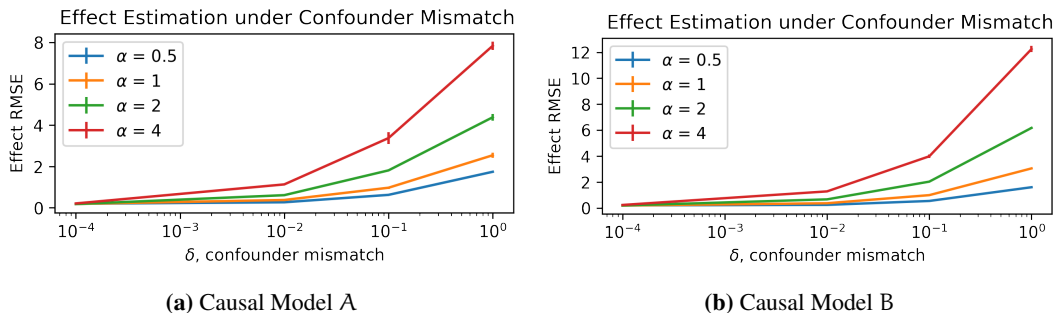

**(a)** Causal Model A             **(b)** Causal Model B

**Figure 6:** RMSE of estimated conditional effect vs. degree of confounder mismatch $\delta$. Error due to conditioning on a mismatched value of the confounder increases with strength of confounding but is mitigated by smoothness of the outcome function.

Third, we investigate the bias induced due to imperfect estimation of the surrogate intervention in LODE for both causal models. We construct surrogate interventions $t'(t^*, h(t_2^*))$ by ensuring there is confounder-value mismatch $h(\tilde{t}) \neq h(t_2^*)$. We do this by interrupting Euler integration when the objective $\mathbb{E}_{t^*, h(t_2^*)}(h(t'(t^*, h(t_2^*))) - h(t_2^*))^2 = \delta^2 > 0$, where the $\mathbb{E}_{t^*, h(t_2^*)}$ is over our evaluation set upon which we estimate conditional effects. For different $\alpha$, we plot in [fig. 6](#) the mean and standard deviation of RMSE of estimated conditional effect over 10 seeds, against different degrees of confounder mismatch, $\delta$. The error due to confounder mismatch is mitigated by small $\alpha$, the Lipschitz-constant of the outcome as a function of $h(t)$. Finally, we consider how step size in Euler integration affects the quality of estimated effects. Large step sizes may result in biased surrogate estimates; this bias is captured in the accumulation error in [section 3.1](#). We focus on the non-linear case in causal model B where gradient errors can accumulate(see [appendix A.3.1](#)). We demonstrate this error in [fig. 5](#) where we plot mean and standard deviation of conditional effect RMSE against the strength of confounding, for different step sizes $\ell$. We do not report results for larger step sizes ($\ell > 2$) because Euler integration diverged for many surrogate estimates.

## 4.2 Effects in Genetics (GWAS)

In this experiment, we explore the associations of genetic factors and Celiac disease. We utilize data from the Wellcome Trust Celiac disease GWAS dataset [8, 6] consisting of individuals with celiac disease, called cases ($n = 3796$), and controls ($n = 8154$). We construct our dataset by filtering from the $\sim 550,000$ SNPs. The only preprocessing in our experiments is linkage disequilibrium pruning of adjacent SNPs (at 0.5 $R^2$) and PLINK [5] quality control. After this, $337,642$ SNPs remain for $11,950$ people. We imputed missing SNPs for each person by sampling from the marginal distribution of that SNP. No further SNP or person was dropped due to missingness. The objective of this experiment is to show that LODE corrects for confounding and recovers SNPs reported in the literature [8, 25, 14, 1]. To this end, after preprocessing, we included in our data 50 SNPs reported in [8, 25, 14, 1] and 1000 randomly sampled from the rest.

We use outcome models and functional confounders $h()$ traditionally employed in the GWAS literature. We choose a linear $h(\tilde{t}) = A^\top \tilde{t}$, where $A$ is a matrix of the right singular vectors of a normalized Genotype matrix, that correspond to the top 10 singular values [19]. The outcome model is selected from logistic Lasso linear models with various regularization strengths, via cross validation within the training data (60% of the dataset). We defer details about the experimental setup to [appendix B](#).

We then use this outcome model in LODE to compute causal effects on the whole filtered dataset. The effects are computed one SNP at a time. First, for each person $\tilde{t}$, create $\tilde{t}_i^1, \tilde{t}_i^0$ which correspond to the $i$th SNP set to 1 and 0 respectively, with all other SNPs same as $\tilde{t}$. Randomly sample a $h(t_2^*)$ from the marginal $p(h(t))$ and, using the outcome model $P_\theta$, compute $\phi(\tilde{t}, i) = \log {}^{P_\theta(y=1 \mid t'(\tilde{t}_i^1, h(t_2^*)))}/_{P_\theta(y=1 \mid t'(\tilde{t}_i^0, h(t_2^*)))}$. The average effect of SNP $i$ is obtained by averaging across all persons: $\sum_{\tilde{t}} \phi(\tilde{t}, i)/N$. Any SNP that beats a specified threshold of effect is deemed relevant to Celiac disease by LODE. We use a $60 - 40\%$ train-test split, and outcome model selection is done via cross-validation within the training set. We did 5-fold cross-validation using just the training set. We use Scikit-learn [18] to fit the outcome models and for cross-validation.

**Results** The best outcome model was a Lasso model, trained with regularization constant 10. We select relevant SNPs by thresholding estimated effects at a magnitude $> 0.1$. From 1050 SNPs (1000 not reported before) LODE returned 31 SNPs, out of which 13 were previously reported as being

associated with Celiac disease [8, 25, 14, 1]. In appendix B.2 we plot the true positive and false negative rates of identifying previously reported SNPs, as a function of the effect threshold.

In table 1, we list a few SNPs that were both deemed relevant by LODE and were reported in existing literature [8, 25, 14, 1], their effects, and their Lasso coefficients. The full list is in table 2 in appendix B. If LODE cannot adjust for confounding, the Lasso coefficients would dictate the effects; 0 coefficient means 0 effect. However, the two pairs of SNPs in table 1 show that the effects estimated by LODE do not rely solely on the Lasso coefficients. For the first pair (rs13151961, rs2237236), the effect is the same but the coefficient of one is 0, while the other is positive. We note that rs2237236 was found to be associated with ulcerative colitis [12, 2], which is an inflammatory bowel disease

| SNP | EFFECT. | COEF. |
|---|---|---|
| rs13151961 | 0.17 | 0.32 |
| rs2237236 | 0.17 | 0.00 |
| rs1738074 | −0.16 | −0.23 |
| rs11221332 | −0.15 | −0.24 |

**Table 1:** A few SNPs previously reported as relevant and recovered by LODE, with estimated effects and Lasso coefficients. LODE produces effect estimates that do not rely purely on the coefficients.

that has been reported to share some common genetic basis with celiac disease [16]. For the second pair, (rs1738074, rs11221332), the magnitude of the effect is smaller for the former, but the coefficient is larger. Thus, LODE adjusts for confounding factors that the outcome model ignored.

## 5  Discussion

When positivity is violated in traditional OBS-CI, not all effects are estimable without further assumptions. In such cases, practitioners have to turn to parametric models to estimate causal effects. However, parametric models can be misspecified when used without underlying causal mechanistic knowledge. We develop a new general setting of observational causal effect estimation called estimation with functional confounders (EFC) where the confounder can be expressed as a function of the data, meaning positivity is violated. Even when positivity is violated, the effects of many functional interventions are estimable. We develop a sufficient condition called functional positivity (F-POSITIVITY) to estimate effects of functional interventions. Such effects could be of independent interest; like the effect of cumulative dosage of a drug instead of joint effects of multiple dosages at different times.

Second, we prove a necessary condition for nonparametric estimation of effects of the full intervention. We propose the C-REDUNDANCY condition, under which, the effect of the full intervention on $\mathbf{t}$ is estimable without parametric restrictions. We develop Level-set Orthogonal Descent Estimation (LODE) that computes surrogate interventions whose effects are estimable and match a conditional effect of interest. Further, we give bounds on errors (theorem 2) induced due to imperfect estimation of the surrogate intervention. Finally, we empirically demonstrate LODE's ability to correct for confounding in both simulated and real data.

**Future.**  A few directions of improvement remain which we elaborate next. First, F-POSITIVITY may not hold for all functions $g(\mathbf{t})$ that we want to intervene on. Instead, one could compute a "projection" $g_\Pi$ to the space of functions that satisfy F-POSITIVITY and inspect the effects defined by $g_\Pi$ instead. A second direction of interest is to let $h(\mathbf{t})$ only account for a part of the confounding, meaning ignorability is violated. This bias could be mitigated under smoothness conditions of the outcome function and its interaction with the degree of violation of ignorability.

Finally, LODE's search strategy is Euler integration, which is equivalent to gradient descent with a fixed step size. Optimization techniques like momentum, rescaling the gradient using an adaptive matrix, and using second order hessian information, speed up gradient descent. However, if there are many local or global minima for $(h(\tilde{\mathbf{t}}) - h(\mathbf{t}_2^*))^2$, such techniques will result in a different solution than Euler integration, which could mean that effect estimates are biased. One extension of LODE would allow for search strategies that use such techniques.

## Broader Impact

Our work mainly applies to causal inference where confounders are specified as functions of observed data, such as in problems in genetics and healthcare. We choose to assess the impact of our work through its applications in these fields. A positive impact of the work is that better estimates of causal effects helps guide treatment for people and aid in understanding biological pathways of diseases. However, in healthcare, data collected in hospitals has biases. If, for instance, a certain demographic of people have more complete data collected about them, then this demographic would have better quality effect estimates, potentially meaning that they receive better treatment. This problem could be characterized by evaluating the positivity of treatment and completeness of confounders in electronic health record data split by demographics.

## Acknowledgements

The authors were partly supported by NIH/NHLBI Award R01HL148248, and by NSF Award 1922658 NRT-HDR: FUTURE Foundations, Translation, and Responsibility for Data Science. The authors would like to thank Xintian Han, Raghav Singhal, Victor Veitch, Fredrik D. Johansson and the reviewers for thoughtful feedback. The authors would also like to thank Mukund Sudarshan and Prof. Sriram Sankararaman for help with running the GWAS experiments.

## Footnotes

[1]We focus on $f$ that generates $y$ from $t, z$. SCMs generally specify the function that generates $t$ from $z$ also.

[2]We also assume no interference [10] (also called Stable Unit Treatment Value Assumption [24]) which means that an individual's outcome does not depend on others' treatment. In EFC, when $\mathbf{t}$ and $\boldsymbol{\eta}$ are sampled IID there is no interference. To see this, note $\forall i, j \ (\mathbf{t}_i, \boldsymbol{\eta}_i) \perp\!\!\!\perp (\mathbf{t}_j, \boldsymbol{\eta}_j) \implies (\mathbf{y}_i, \mathbf{t}_i) \perp\!\!\!\perp (\mathbf{y}_j, \mathbf{t}_j) \implies \mathbf{y}_i \perp\!\!\!\perp \mathbf{t}_j$.

[3]Intervening on $g(\mathbf{t})$ can be interpreted as making a *soft intervention* [9, 7] of $\mathbf{t}$ to $p(\mathbf{t} \mid \mathbf{z}, g(\mathbf{t}) = g(\tilde{\mathbf{t}}))$.

[4]If $f$ transforms its first argument $\tilde{t}$ into $h(\tilde{t})$ as one amongst many different computations, the chain rule implies $\nabla_{\tilde{t}} f(\tilde{t}, h(t_2^*))^\top \nabla_{\tilde{t}} h(\tilde{t})$ has a term $\|\nabla_{\tilde{t}} h(\tilde{t})\|^2$ which is non-zero in general.

[5]We ignore noise in the outcome for ease of exposition.

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
