[Supplementary Material]

# A Theoretical details

## A.1 A note about the assumptions

**Note about the assumptions** In theorem 1, assumption 1 consists of three parts that can all be validated on observed data: 1) that the gradient flow converges, 2) that the confounder value of the surrogate matches the confounder value whose effect is of interest, and 3) that the surrogate intervention lies in the support of the pre-outcome variables. Assumption 2 is required for expectations and their gradients to exist and be finite. In theorem 2, assumption 1 requires a consistent estimator of $\mathbb{E}[y \mid t]$, which can be provided with regression. Assumption 3 lists regularity conditions which help control how the surrogate estimation error propagates to the effect error.

## A.2 Proof of Theorem 1

We restate the theorem for completeness:

**Theorem 1.** *Assume* C-REDUNDANCY *holds. Assuming the following:*

1. *Let $t'(t^*, h(t_2^*))$ be the limiting solution to the gradient flow equation $\frac{d\tilde{t}(s)}{ds} = -\nabla_{\tilde{t}}(h(\tilde{t}(s)) - h(t_2^*))^2$, initialized at $\tilde{t}(0) = t^*$; i.e. $t'(t^*, h(t_2^*)) = \lim_{s \to \infty} \tilde{t}(s)$. Further, let $h(t'(t^*, h(t_2^*))) = h(t_2^*)$ and $t'(t^*, h(t_2^*)) \in supp(t)$.*

2. *$f(\tilde{t}, h(\tilde{t}), \eta)$ and $h(\tilde{t})$ as functions of $\tilde{t}, h(\tilde{t})$ are continuous and differentiable and the derivatives exist for all $\tilde{t}, \eta$. Let $\nabla_{\tilde{t}} f(\tilde{t}, h(\tilde{t}), \eta)$ exist and be bounded and integrable w.r.t. the probability measure corresponding to $p(\eta)$, for all values of $\tilde{t}$ and $h(\tilde{t})$.*

*Then the conditional effect (and therefore the average effect) is identified:*

$$\phi(t^*, h(t_2^*)) = \phi\left(t'(t^*, h(t_2^*)), h(t'(t^*, h(t_2^*)))\right) = \mathbb{E}\left[y \mid t = t'(t^*, h(t_2^*))\right] \qquad (10)$$

*Proof.* Recall definition of conditional effect $\phi(\tilde{t}, h(\tilde{t}_2)) = \mathbb{E}_\eta f(\tilde{t}, h(\tilde{t}_2), \eta)$. Recall $\nabla_{\tilde{t}}$ is the gradient with respect to the first argument of f, that is $\tilde{t}$. First, by assumption 2, $\mathbb{E}$ and $\nabla$ commute, under the dominated convergence theorem. Then, by C-REDUNDANCY

$$\nabla_{\tilde{t}}\phi(\tilde{t}, h(t^*))^\mathsf{T}\nabla_{\tilde{t}}h(\tilde{t}) = \nabla_{\tilde{t}}\mathbb{E}_\eta f(\tilde{t}, h(t^*), \eta)^\mathsf{T}\nabla_{\tilde{t}}h(\tilde{t}) = \mathbb{E}_\eta[\nabla_{\tilde{t}}f(\tilde{t}, h(t^*), \eta)^\mathsf{T}\nabla_{\tilde{t}}h(\tilde{t})] = 0.$$

Now consider the gradient flow equation $d\tilde{t}(s)/ds = -\nabla_{\tilde{t}}(h(\tilde{t}) - h(t_2^*))^2$. We refer to the gradient evaluated at $\tilde{t}$ as $\Delta\tilde{t} = -\nabla_{\tilde{t}}(h(\tilde{t}) - h(t_2^*))^2 = -2(h(\tilde{t}) - h(t_2^*))\nabla_{\tilde{t}}h(\tilde{t})$. We will express $\phi(t'(t^*, h(t_2^*)), h(t_2^*))$ as defined by the starting point $\phi(t^*, h(t_2^*))$ and the gradient flow equation.

Let the solution path to the gradient flow equation be C with $t^*, t'(t^*, h(t_2^*))$ being the starting and ending points respectively. By the Gradient Theorem [26], we have that $\phi(t^*, h(t_2^*))$ and $\phi(t'(t^*, h(t_2^*)), h(t_2^*))$ are related via the line integral over C:

$$\int_C \nabla_{\tilde{t}}\phi(\tilde{t}, h(t_2^*)) \cdot d\tilde{t} = \phi(t'(t^*, h(t_2^*)), h(t_2^*)) - \phi(\tilde{t}, h(t_2^*))$$

Let $\tilde{t}(s)$ be a parametrization of solution path C by the scalar time $s \in [0, \infty)$. Now, to obtain the value of $\phi(\tilde{t}, h(t_2^*))$, we will compute the line integral over the vector field defined by $\nabla_{\tilde{t}}\phi(\tilde{t}, h(t_2^*))$, which exists by assumption 2 in theorem 1, evaluated along the path C defined by $\Delta\tilde{t}(s)$:

$$
\begin{aligned}
\phi(t'(t^*, h(t_2^*)), h(t_2^*)) &= \phi(t^*, h(t_2^*)) + \int_C \nabla_{\tilde{t}}\phi(\tilde{t}, h(t_2^*)) \cdot d\tilde{t} \\
&= \phi(t^*, h(t_2^*)) + \int_0^\infty \nabla_{\tilde{t}}\phi(\tilde{t}(s), h(t_2^*))^\mathsf{T}\frac{d\tilde{t}(s)}{ds} \, ds \\
&= \phi(t^*, h(t_2^*)) + \int_0^\infty \nabla_{\tilde{t}}\phi(\tilde{t}(s), h(t_2^*))^\mathsf{T}\Delta\tilde{t}(s) \, ds \\
&= \phi(t^*, h(t_2^*)) \\
&\quad + \int_0^\infty -2((h(\tilde{t}(s)) - h(t_2^*)))\,\nabla_{\tilde{t}}\phi(\tilde{t}(s), h(t_2^*))^\mathsf{T}\nabla_{\tilde{t}}h(\tilde{t}(s)) \, ds \\
&= \phi(t^*, h(t_2^*)) + 0 \qquad \{\text{by C-REDUNDANCY}\}
\end{aligned}
$$

$$(11)$$

Finally, by assumption 1 in theorem 1, $h(t'(t^*, h(t_2^*))) = h(t_2^*)$, and so

$$\varphi(t^*, h(t_2^*)) = \varphi(t'(t^*, h(t_2^*)), h(t_2^*)) = \varphi(t'(t^*, h(t_2^*)), h(t'(t^*, h(t_2^*)))) \quad (12)$$

For clarity, the same equation, but using $t'$ and suppressing dependence on $t^*, h(t_2^*)$:

$$\varphi(t^*, h(t_2^*)) = \varphi(t', h(t_2^*)) = \varphi(t', h(t')) \quad (13)$$

Under the causal model for EFC, the outcome $y = f(t, h(t), \eta)$. Then, $\forall \tilde{t} \in \mathrm{supp}(p(t))$,

$$\mathbb{E}[y \,|\, t = \tilde{t}] = \mathbb{E}_\eta[f(\tilde{t}, h(\tilde{t}), \eta)] = \varphi(\tilde{t}, h(\tilde{t})). \quad (14)$$

Using that $t'(t^*, t_2^*) \in \mathrm{supp}(p(t))$ and eqs. (13) and (14), the conditional effect is identified

$$\begin{aligned}
\varphi(t^*, h(t_2^*)) &= \varphi(t'(t^*, h(t_2^*)), h(t'(t^*, h(t_2^*)))) \\
&= \mathbb{E}[y \,|\, t = t'(t^*, h(t_2^*))]
\end{aligned} \quad (15)$$

Thus, the conditional effect, and consequently the average effect, are identified as $\mathbb{E}[y \,|\, t'(t^*, h(t_2^*))]$ and $\tau(t^*) = \mathbb{E}_{h(t)}\mathbb{E}[y \,|\, t'(t^*, h(t))]$ respectively. $\qquad\square$

**Note about convergence of gradient flow** Any ODE's solution, if it exists and converges, converges to an $\omega$-limit set [27]. An $\omega$-limit set is nonempty when the solution path lies entirely in a closed and bounded set and can consist of limit cycles, equilibrium points, or neither [13, 27]. A gradient flow equation $\mathrm{d}\tilde{t}(s)/\mathrm{d}s = -\nabla h(\tilde{t})$ (also called a gradient system) has the special property that its $\omega$-limit set only consists of critical points of $h(\tilde{t})$; critical points of $h(\tilde{t})$ are also equilibrium points of the gradient flow equation [13]. Further, if $\nabla h(\tilde{t})$ exists and is bounded and $h(\tilde{t})$ has bounded sublevel sets ($\{\tilde{t} : h(\tilde{t}) \leqslant c\}$), then the solution to the gradient flow equation will entirely lie within a bounded set. This is because along the solution path, $h(\tilde{t}(s))$ always decreases meaning that the solution will remain in any sublevel set it started in. Thus, if $h(\tilde{t})$ has bounded sublevel sets, the solution of the gradient flow equation will converge only to critical points of $h(\tilde{t})$.

## A.3 Estimation error in LODE

**Theorem 2.** *Consider the conditional effect $\varphi(t^*, h(t_2^*))$. Let $\hat{t}(t^*, h(t_2^*))$ be the estimate of the surrogate intervention computed by* LODE*, computed via Euler integration of the gradient flow $\frac{\mathrm{d}\tilde{t}(s)}{\mathrm{d}s} = -\nabla_{\tilde{t}}(h(\tilde{t}(s)) - h(t_2^*))^2$, initialized at $\tilde{t}(0) = t^*$. Assume the true surrogate $t'(t^*, h(t_2^*))$ exists and is the limiting solution to the gradient flow equation.*

1. *Let the finite sample estimator of $\mathbb{E}[y \,|\, t = \tilde{t}]$ be $\hat{f}(\tilde{t})$. Let the error for all $\tilde{t}$ be bounded, $|\hat{f}(\tilde{t}) - \mathbb{E}[y \,|\, t = \tilde{t}]| \leqslant c(N)$, where $N$ is the sample size and $\lim_{N \to \infty} c(N) = 0$.*

2. *Assume $K$ Euler integrator steps were taken to find the surrogate estimate $\hat{t}(t^*, h(t_2^*))$, each of size $\ell$. Let the maximum confounder mismatch be $\max_{i \leqslant K}(h(\tilde{t}_i) - h(t_2^*))^2 = M$.*

3. *Let $L_{z,\tilde{t}}$ be the Lipschitz-constant of $\varphi(\tilde{t}, h(\tilde{t}_2))$ as a function of $h(\tilde{t}_2)$, for fixed $\tilde{t}$. Let $L_e$ be the Lipschitz-constant of $\mathbb{E}[y \,|\, t = \tilde{t}] = \varphi(\tilde{t}, h(\tilde{t}))$ as a function of $\tilde{t}$. Assume $h$ has a gradient with bounded norm, $\|\nabla h(\tilde{t})\|_2 < L_h$. Assume $f$'s Hessian has bounded eigenvalues: $\forall \tilde{t}, \tilde{t}_2, \|\nabla_{\tilde{t}}^2 \varphi(\tilde{t}, h(\tilde{t}_2))\|_2 \leqslant \sigma_{H\varphi}$.*

*The conditional effect estimate error, $\xi(t^*, h(t_2^*)) = |\hat{f}(\tilde{t}) - \varphi(t^*, h(t_2^*))|$, is upper bounded by:*

$$c(N) + \min\left(L_e\|t' - \hat{t}\|_2, \ 2K\ell^2\left(\mathcal{O}(\ell) + M\sigma_{H\varphi}L_h^2\right) + L_{z,\hat{t}}\|h(\hat{t}) - h(t_2^*)\|_2\right) \quad (16)$$

*Proof.* (of Theorem 2) Recall the definition of conditional effect : $\varphi(\tilde{t}, h(\tilde{t}_2)) = \mathbb{E}_\eta f(\tilde{t}, h(\tilde{t}_2), \eta)$.

LODE's estimate of the conditional effect is $\hat{f}(\hat{t}(t^*, h(t_2^*)))$. We will suppress notation for dependence on $t^*, h(t_2^*)$, and use $t'$ and $\hat{t}$ to refer to the true surrogate intervention and the estimated surrogate interventions respectively. Note $\hat{f}$ is the estimate of the conditional expectation $\mathbb{E}[y \,|\, t = \tilde{t}]$, learned from $N$ samples. We first bound the error by splitting into two parts and bounding each separately:

$$\begin{aligned}
|\xi(t^*, h(t_2^*))| = |\hat{f}(\hat{t}) - \varphi(t^*, h(t_2^*))| \\
\leqslant |\hat{f}(\hat{t}) - \varphi(\hat{t}, h(\hat{t}))| + |\varphi(\hat{t}, h(\hat{t})) - \varphi(t^*, h(t_2^*))| \\
\leqslant c(N) + |\varphi(\hat{t}, h(\hat{t})) - \varphi(t^*, h(t_2^*))| \\
\leqslant |\varphi(\hat{t}, h(\hat{t})) - \varphi(\hat{t}, h(t_2^*))| + |\varphi(\hat{t}, h(t_2^*)) - \varphi(t^*, h(t_2^*))| + c(N)
\end{aligned}$$

The first term is bounded via the Lipschitz-ness of $\phi$ as a function of $h(\tilde{t})$ with fixed first argument $\tilde{t} = \hat{t}$.

$$|\phi(\hat{t}, h(\hat{t})) - \phi(\hat{t}, h(t_2^*))| \leqslant L_{z,\hat{t}}|h(\hat{t}) - h(t_2^*)|$$

We now bound the remaining term. Recall that LODE's computation of the surrogate intervention involved $K$ gradient steps, each of size $\ell$. We work with a constant step-size but the analysis can be generalized to a non-uniform step size. Indexing steps with $i$, let $d_i = h(\tilde{t}_i) - h(t_2^*)$ be the confounder mismatch error at the $i$th iterate. Then note that $\hat{t} = t^* - \ell \sum_{i=0}^{K-1} 2d_i \nabla_{\tilde{t}} h(\tilde{t}_i)$. We can use this to bound the error $\phi(\hat{t}, h(t_2^*)) - \phi(t^*, h(t_2^*))$. With $\tilde{t}_K = \hat{t}$ and $\tilde{t}_0 = t^*$, we proceed by expressing the error as a telescoping sum and using the Taylor expansion for $\phi(\tilde{t}, h(t_2^*))$ in terms of the the first argument $\tilde{t}$.

$$\phi(\hat{t}, h(t_2^*)) - \phi(t^*, h(t_2^*)) = \sum_{i=0}^{K-1} \phi(\tilde{t}_{i+1}, h(t_2^*)) - \phi(\tilde{t}_i, h(t_2^*)) \tag{17}$$

$$= \sum_{i=0}^{K-1} \nabla_{\tilde{t}} \phi(\tilde{t}_i, h(t_2^*))^\top (\tilde{t}_{i+1} - \tilde{t}_i) \tag{18}$$

$$+ \frac{1}{2}(\tilde{t}_{i+1} - \tilde{t}_i)^\top \nabla_{\tilde{t}}^2 \phi(\tilde{t}_i, h(t_2^*))(\tilde{t}_{i+1} - \tilde{t}_i) + \mathcal{O}(\|\tilde{t}_{i+1} - \tilde{t}_i\|_2^3) \tag{19}$$

$$= \sum_{i=0}^{K-1} 2\ell d_i \nabla_{\tilde{t}} \phi(\tilde{t}_i, h(t_2^*))^\top \nabla_{\tilde{t}} h(\tilde{t}_i) + 2(\ell d_i)^2 \nabla_{\tilde{t}} h(\tilde{t}_i)^\top \nabla_{\tilde{t}}^2 \phi(\tilde{t}_i, h(t_2^*)) \nabla_{\tilde{t}} h(\tilde{t}_i) + \mathcal{O}(\ell^3) \tag{20}$$

$$= \sum_{i=0}^{K-1} 0 + 2(\ell d_i)^2 \nabla_{\tilde{t}} h(\tilde{t}_i)^\top \nabla_{\tilde{t}}^2 \phi(\tilde{t}_i, h(t_2^*)) \nabla_{\tilde{t}} h(\tilde{t}_i) + \mathcal{O}(\ell^3) \tag{21}$$

$$= \mathcal{O}(K\ell^3) + \sum_{i=0}^{K-1} 2(\ell d_i)^2 \nabla_{\tilde{t}} h(\tilde{t}_i)^\top \nabla_{\tilde{t}}^2 \phi(\tilde{t}_i, h(t_2^*)) \nabla_{\tilde{t}} h(\tilde{t}_i) \tag{22}$$

$$\leqslant \mathcal{O}(K\ell^3) + \sum_{i=0}^{K-1} 2(\ell(h(\tilde{t}_i) - h(t_2^*)))^2 \left| \nabla_{\tilde{t}} h(\tilde{t}_i)^\top \nabla_{\tilde{t}}^2 \phi(\tilde{t}_i, h(t_2^*)) \nabla_{\tilde{t}} h(\tilde{t}_i) \right| \tag{23}$$

$$\leqslant \mathcal{O}(K\ell^3) + \sum_{i=0}^{K-1} 2\ell^2 M \left| \nabla_{\tilde{t}} h(\tilde{t}_i)^\top \nabla_{\tilde{t}}^2 \phi(\tilde{t}_i, h(t_2^*)) \nabla_{\tilde{t}} h(\tilde{t}_i) \right| \tag{24}$$

$$\leqslant \mathcal{O}(K\ell^3) + \sum_{i=0}^{K-1} 2\ell^2 M \sigma_{H\phi} \|\nabla_{\tilde{t}} h(\tilde{t}_i)\|_2^2 \tag{25}$$

$$\leqslant \mathcal{O}(K\ell^3) + \sum_{i=0}^{K-1} 2\ell^2 M \sigma_{H\phi} L_h^2 \tag{26}$$

$$= 2K\ell^2 \left( \mathcal{O}(\ell) + M \sigma_{H\phi} L_h^2 \right), \tag{27}$$

where the inequalities follow by the maximum value of $(h(\tilde{t}_i) - h(t_2^*))^2$, bounded eigenvalues of the Hessian of $\phi$ and the Lipschitz-ness of $h(\tilde{t})$.

Another way we bound the error is via the Lipschitz constant of the conditional expectation as a function of $\tilde{t}$. Recall this is $L_e$. An alternate bound on the error is as follows:

$$|\phi(\hat{t}, h(\hat{t})) - \phi(t^*, h(t_2^*))| = |\phi(\hat{t}, h(\hat{t})) - \phi(t', h(t'))| \leqslant L_e \|t' - \hat{t}\|_2$$

The bound follows:

$$|\xi(\tilde{t}, h(t_2^*))| \leqslant c(N) + \min\left( L_e \|t' - \hat{t}\|_2, \quad 2K\ell^2 \left( \mathcal{O}(\ell) + M \sigma_{H\phi} L_h^2 \right) + L_{z,\hat{t}} \|h(\hat{t}) - h(t_2^*)\|_2 \right)$$

$\square$

### A.3.1 A note on linear confounder functions and LODE

In the proof above, the error in Euler integration accumulates due to terms like this one: $\nabla_{\tilde{t}}^{\top} h(\tilde{t}) \nabla_{\tilde{t}}^2 f(\tilde{t}, h(t^*), \eta) \nabla_{\tilde{t}} h(\tilde{t})$. For a linear confounder function that satisfies $\nabla_{\tilde{t}} h(\tilde{t}) = \beta$, such terms can be expressed as $\beta^{\top} \nabla_{\tilde{t}} (\nabla_{\tilde{t}} f(\tilde{t}, h(t^*), \eta)^{\top} \beta) = \beta^{\top} \nabla_{\tilde{t}}(0) = 0$ under C-REDUNDANCY. Thus, such error does not accumulate even with large step sizes.

Further, note that the gradient flow equation in LODE for the causal model $A$ in section 4 is a linear ODE whose solution has a closed form expression and one can estimate the surrogate without numerical integration [27].

### A.4 Proof of sufficiency of Effect Connectivity

**Theorem 3.** *Under Effect Connectivity, eq. (9), any surrogate intervention* $t'(t^*, h(t_2^*)) \in supp(t)$.

*Proof.* Recall $\phi(\tilde{t}, h(\tilde{t})) = \mathbb{E}_{\eta} f(\tilde{t}, h(\tilde{t}), \eta)$. We have $\forall t^* \in \text{supp}(p(t))$:

$$p(h(t) = h(t_2^*)) > 0 \implies p(\phi(t, h(t)) = \phi(t^*, h(t_2^*)) \mid h(t) = h(t_2^*)) > 0.$$

This implies $\exists t' \in \text{supp}(t), \phi(t', h(t_2^*)) = \phi(t^*, h(t_2^*)),$ s.t. $h(t') = h(t_2^*)$.

Then, $\phi(t^*, h(t_2^*)) = \phi(t', h(t_2^*)) = \phi(t', h(t')) = \mathbb{E}[y \mid t = t']$. $\qquad \square$

### A.5 Necessity of Effect Connectivity for Nonparametric effect estimation in EFC

**Theorem 4.** *Effect Connectivity is necessary for nonparametric effect estimation in* EFC.

*Proof.* (Proof of Theorem 4) Let the outcome be $y = f(t, h(t))$. Recall the joint distribution $p(t, y)$ and let $h(t)$ be the confounder. Let Effect Connectivity be violated, i.e. there exists a non-measure-zero subset $B \in \text{supp}(t) \times \text{supp}(h(t))$ such that [6]:

$$\forall \tilde{t}, h(\tilde{t}_2) \in B, \qquad p(f(t, h(t)) = f(\tilde{t}, h(\tilde{t}_2)) \mid h(t) = h(\tilde{t}_2)) = 0.$$

Now, we construct a new outcome $y_2 = f_2(t, h(t))$ and show the conditional effects for this new outcome are different from the one defined by f on $\forall(\tilde{t}, h(\tilde{t}_2)) \in B$. Let

$$f_2(\tilde{t}, h(\tilde{t}_2)) = f(\tilde{t}, h(\tilde{t}_2)) + 10 * 1((\tilde{t}, h(\tilde{t}_2)) \in B)|.$$

We have $f_2(\tilde{t}, h(\tilde{t})) = f(\tilde{t}, h(\tilde{t})) \forall \tilde{t} \in \text{supp}(t)$, as the additional term in $f_2$ is only present for $(\tilde{t}, h(\tilde{t}_2)) \in B$; this follows from the fact that $\forall \tilde{t} \in \text{supp}(t), (\tilde{t}, h(\tilde{t})) \notin B$ as

$$p[f(t, h(t)) = f(\tilde{t}, h(\tilde{t})) \mid h(t) = h(\tilde{t})] = p[f(t, h(t)) = f(\tilde{t}, h(\tilde{t}))] > 0.$$

Thus, $p(y, t) =^{d} p(y_2, t)$ are equal in distribution since $B \cap \text{supp}(t, h(t)) = \emptyset$. This means that the conditional effects are different for the outcomes $y, y_2$ for all $(\tilde{t}, h(\tilde{t}_2)) \in B$:

$$\mathbb{E}[y \mid do(t = \tilde{t}), h(t) = h(\tilde{t}_2)] \neq \mathbb{E}[y_2 \mid do(t = \tilde{t}), h(t) = h(\tilde{t}_2)]$$

Therefore, for causal models that violates Effect Connectivity, there exist observationally equivalent causal models with different causal effects. Thus, nonparametric effect estimation is impossible. Thus, Effect Connectivity is required for EFC. $\qquad \square$

### A.6 Algorithmic details

We give in algorithm 1 pseudocode for LODE.

**Extensions of LODE** Consider that we have access to $m(h(t))$ for some bijective differentiable function $m(\cdot)$, instead of $h(t)$. The orthogonality in C-REDUNDANCY holds $\nabla_{\tilde{t}} f(\tilde{t}, h(\tilde{t}_2), \eta)^{\top} \nabla_{\tilde{t}} m(h(\tilde{t})) = m'(h(\tilde{t})) \nabla_{\tilde{t}} f(\tilde{t}, h(\tilde{t}_2), \eta)^{\top} \nabla_{\tilde{t}} h(\tilde{t}) = 0$. Then, using $m(h(\tilde{t}))$ to compute the surrogate $t'(t^*, h(t_2^*))$, LODE would estimate valid effects. Similarly, LODE can estimate the effect on any differentiable transformation of the outcome $m(y)$, because $\nabla_{\tilde{t}} m(y_{\tilde{t}})^{\top} \nabla_{\tilde{t}} h(\tilde{t}) = m'(y_{\tilde{t}}) \nabla_{\tilde{t}} f(\tilde{t}, h(\tilde{t}_2), \eta)^{\top} \nabla_{\tilde{t}} h(\tilde{t}) = 0$ holds.

**Algorithm 1:** LODE for $do(t = t^*)$

---

**Input:** Functional confounder $h(t)$; tolerance $\epsilon$
**Output:** Conditional effects of $t^*, h(t_2^*)$

1. Regress $y$ on $t$ and compute $\hat{f}() := \arg\min_{u \in \mathcal{F}} \mathbb{E}_{y,t}(y - u(t))^2$.
2. To estimate effects of $t^*, h(t_2^*)$, compute the surrogate intervention $t'(t^*, h(t_2^*))$ by Euler integrating the gradient flow equation, initialized at $\tilde{t} = t^*$, until $(h(\tilde{t}_s) - h(t_2^*))^2 < \epsilon$.

$$\frac{d\tilde{t}(s)}{ds} = \nabla_{\tilde{t}}(h(\tilde{t}_s) - h(t_2^*))^2,$$

3. Return $\hat{f}(t'(t^*, h(t_2^*)))$;

---

## B  Experimental Details

### B.1  Functional confounders in GWAS

Here, we show how $h(t) = At$ and $A$ reflect the traditional PCA based adjustment in GWAS. Recall population structure acts as a confounder in GWAS. Price et al. [19] demonstrated that using the principal components of the normalized genetic relationships matrix adjusts for confounding due to population structure in GWAS. Let the genotype matrix be $G$ with people as rows and SNPs as columns, such that each element is one of $0, 1/2, 1$, where $1/2$ and $1$ refer to one and two copies of the allele respectively at the position of the SNP. With $p_s$ as the allele frequency at SNP $s$ [28], $\Phi$ is the genetic relationship matrix whose elements are defined as $\Phi_{i,j} = \frac{1}{S}\sum_{s=1}^{S} (G_{i,s} - p_s)(G_{j,s} - p_s)/p_s(1 - p_s)$. Then, Price et al. [19] compute the top $K$ (10 suggested) principal components of $\Phi$ to use as the axes of variation due to the population structure. The eigenvectors of $\Phi$ are the left eigenvectors of $\hat{G}$ such that $\Phi = \hat{G}\hat{G}^\top$ which capture independent axes of variation of individuals.

Price et al. [19] exploit the idea that if a SNP aligns with some of the axes of variation, this is due to the population structure. These axes of variation are the top $K$ eigenvectors $U$ of $\phi = \hat{G}\hat{G}^\top \approx U\Lambda U^\top$, where $U \in \mathbb{R}^{N \times K}$, $\Phi \in \mathbb{R}^{N \times N}$ and $\Lambda \in \mathbb{R}^{K \times K}$. Here, $U$ are also the left singular vectors of $\hat{G} \approx U\Sigma V^\top$ where $\Sigma \in \mathbb{R}^{K \times K}$ is diagonal, and $V \in \mathbb{R}^{S \times K}$. We use $\approx$ to denote that the chosen $K$ eigenvectors explain the variation due to population structure; what remains are random mutations.

Let the $s$th SNP be $\hat{G}_{\cdot,s} \in \mathbb{R}^N$, which is a column in $\hat{G}$. In Price et al. [19], population structure in the $s$th SNP is captured in $\hat{G}_{\cdot,s}^\top U$. In words, projecting the SNP $\hat{G}_{\cdot,s}$ onto the axes of variation in individuals gives the population structure between $s$th SNP and the outcome. This projection $\hat{G}_{\cdot,s}^\top U$ is a row of $\hat{G}^\top U \in \mathbb{R}^{S \times K}$. In turn, $\hat{G}^\top U \in \mathbb{R}^{S \times K}$ is the population structure in all SNPs. Projecting this population structure onto the genotype of an individual gives the confounding due to population structure amongst the SNPs present in the genotype. With $G_{j,\cdot} \in \{0, 1/2, 1\}^S$ as the genotype for an individual $j$, this projection is $((\hat{G}^\top U)^\top G_{j,\cdot})$. However, $\hat{G} \approx U\Sigma V^\top$ implies that $\hat{G}^\top U \approx V\Sigma$. Reflecting this, $h(t) = \Sigma V^\top t$ is the functional confounder for an individual $t$.

## B.2 Expanded results

In [table 2](#), we list the 13 SNPs recovered by LODE, that have been previously reported as relevant to Celiac disease. In [fig. 7](#), we plot the true positive and false negative rate amongst SNPs deemed relevant by LODE. The ground truth here are the SNPs reported associated with celiac disease in prior literature.

**Figure 7:** True positive vs. False negative rate as we vary the threshold on average effects, that determines which SNPs LODE deems relevant to the outcome.

| SNP | EFFECT | LASSO COEF. |
|---|---|---|
| rs3748816 | 0.12 | 0.20 |
| rs10903122 | 0.10 | 0.17 |
| rs2816316 | 0.11 | 0.20 |
| rs13151961 | 0.17 | 0.32 |
| rs2237236 | 0.17 | 0.00 |
| rs12928822 | 0.14 | 0.29 |
| rs2187668 | −0.70 | −2.37 |
| rs2327832 | −0.12 | −0.20 |
| rs1738074 | −0.16 | −0.23 |
| rs11221332 | −0.15 | −0.24 |
| rs653178 | −0.13 | −0.21 |
| rs4899260 | −0.12 | −0.19 |
| rs17810546 | −0.12 | −0.20 |

**Table 2:** Full list of SNPs previously reported as relevant that were recovered by LODE, and their estimated effects and Lasso coefficients for SNPs. The effect threshold here is 0.1.

## Footnotes

[6] Non-zero w.r.t. the product measure over $\text{supp}(t) \times \text{supp}(h(t))$ due to p.