[Reviews · NeurIPS 2020]

Review 1

Summary and Contributions: This paper addresses the problem of causal inference when the true confounder value is a function of the observed non-outcome variables. The authors mention that under this setting, the positivity assumption i s violated; which as a result, causal inference is impossible in general. The contribution of this work is investigating two scenarios in which the causal effects are estimable.

Strengths: Causal inference is quite relevant to the NeurIPS community.

Weaknesses: Did not understand well enough to comment.

Correctness: Unfortunately, I could not understand the central claim of this paper: that if confounders are a function of the non-outcome variables, then positivity is violated in general. The authors explain in lines 92-93 that positivity does not hold when z is not equal to h(t). However, this never happens because z=h(t) by definition. I am confused ...

Clarity: I am a researcher working in causal inference, so I’m familiar with many concepts that the paper discusses. Yet, the paper was too difficult for me to understand. Too many inline math often interrupted the flow. Additionally, I think the readability of the submission could be improved by having a running example throughout the manuscript. Please consider re-writing your paper.

Relation to Prior Work: Yes, the prior work and their relationship to the current work is clearly discussed in the related work section.

Reproducibility: No

Additional Feedback: Please provide a response to my comment in the “correctness” section. ===== after rebuttal ===== I appreciate the authors's explanation on why the positivity is violated and that they have added this to their paper. However, I still think that the text is too hard to follow for a general NeurIPS audience though; which is why I have updated my score to "above threshold".


Review 2

Summary and Contributions: This paper proposes a novel confounder model where the value of the confounder can be represented as a function of all non-outcome variables. - By investigating functional interventions, authors developed a sufficient condition to estimate their effects. - Considering intervening on all non-outcome variables, authors developed a sufficient condition to estimate the effect of the full intervention.

Strengths: - The theoretical model is well motivated by a concrete example. - Theoretical results are clearly presented and justified.

Weaknesses: - Not enough explanation on the theoretical assumptions. - No related work compared in the empirical evaluation.

Correctness: The logic is clear and makes sense to me

Clarity: This paper is well written.

Relation to Prior Work: Not clearly discussed.

Reproducibility: Yes

Additional Feedback: - The main results (Theorem 1 and 2) are based on several assumptions. It would be better to explain them briefly to give readers ideas about how strong they are and when they would hold. - This paper does not mention any other confounding models. I was wondering whether there exist similar works? If yes, it would be better to compare them in the experiments or explain why they were not compared. - The current manuscript only mentioned related work in the field of the motivating example (i.e., GWAS). It would be better to also review some similar confounding causal models in the introduction.


Review 3

Summary and Contributions: The authors present a way to estimate causal effects when there is confounding present that can be expressed as a function of the observed variables. This type of confounding typically violates the positivity assumption and is present in e.g. GWAS studies. They establish a sufficient condition (F-positivity) for so-called functional interventions in this setting, as well as a necessary condition (C-redundancy) for non-parametric estimation of the full intervention. Finally they introduce and implement a method called LODE (Level-set order dependent estimation) that uses so-called surrogate interventions that can be estimated and match a conditional effect of interest, thereby effectively adjusting for the confounding. The effectivity of the approach is demonstrated on synthetic data as well as a real-world GWAS example on Celiac disease.

Strengths: Very good paper on an importnat and challenging, albeit somewhat niche, causal inference problem that occurs regularly in the context of GWAS analysis. Good mix of conceptual abstraction, solid mathematical rigour and technical implementation, with some interesting necessary and sufficient conditions, as well as highly nontrivial bounds on resulting conditional effect estimates.

Weaknesses: Its strength is perhaps also its weakness: the approach is technically and conceptually challenging, and covers a very specific causal inference problem requiring very specific case knowledge. To be precise: for when we want to estimate a causal effect, but we know that there is still a confounding component, but we also know that there is a functional/deterministic relation between that confounding component and other observable variables, and we also know the functional form of that relation, and we even know that this is also the only confounding component that remains. That is very specific, however, it happens to be a reasonable situation in GWAS analysis, and so it is more than just an exotic theoretical tour de force. But even then I doubt that many practitioners will be able to easily apply the method, and so there is still a danger that in practice the method becomes a case of a solution that was too difficult for the problem.

Correctness: As far as I can tell the claims and results are correct, although I have to admit that I could not follow all steps in detail, and only skimmed the proofs in the supplement. Experimental evaluation is relevant and to the point. Only caveat is the closing claim in the GWAS example in 4.2 ‘Thus LODE adjusts for confounding factors that the outcome model ignored’: I think they only show that we do indeed get a different result, but whether that actually corresponds to ‘correctly adjusting for confounding factors’ has not been proven. Although I am happy to give the authors the benefit of the doubt here.

Clarity: The paper is well-written, but despite the clearly best efforts of the authors it remains highly abstract and technical, which makes it a tough read for anyone not very familiar with the problem already. I guess that this is unavoidable given the material involved, although the paper could benefit from a few more concrete examples. (in particular for the 'surrogate interventions')

Relation to Prior Work: Yes: current work is clearly novel.

Reproducibility: Yes

Additional Feedback: In short: good paper on difficult problem. Quite challenging, but clearly deserves to be in the conference. => accept minor comment: l.65, l.91: typo PCA ’principle’ => ‘principal’

[Author Response · NeurIPS 2020]

We thank the reviewers for their feedback. We're glad the reviewers see our work as tackling an important and
challenging problem and found our theory interesting and well-motivated by a real example. We appreciate that our
presentation is abstract and technical, which made it difficult for R1 to follow, but as R3 pointed out is despite best
efforts, the abstract nature of the presentation is potentially unavoidable given the material.

**[R2: Strength of assumptions for theorem 1 and theorem 2]** Both theorems rely on causal redundancy. This is an
untestable assumption. However, causal redundancy is plausible in settings like GWAS where the population structure
and the effects are orthogonal. See PCA correction, Price et al. and appendix B.1 of our paper for a discussion.

In theorem 1, assumption 1 has three parts: 1) that the gradient flow converges, 2) that the confounder value of the
surrogate matches the confounder value whose effect is of interest, and 3) that the surrogate intervention lies in the
support of the pre-outcome variables. Further, all parts can be validated on observed data. We have expanded the
discussion about assumption 1. Assumption 2 is a standard technical condition required for expectations and their
gradients to exist and be finite.

Coming to theorem 2, assumption 1 requires a consistent estimator of $\mathbb{E}[\mathbf{y} \mid \mathbf{t}]$, which can be provided with regression.
Assumption 2 simply defines how the surrogate estimate $\hat{t}$ is obtained. Assumption 3 is a set of standard regularity
assumptions which help control how the surrogate estimation error propagates to the effect error, like bounded Lipschitz-
constant on the outcome and confounder functions and bounded spectral norm on the Hessian. We have added this
discussion to the paper.

**[R2: Comparison with prior work]** Thank you for this question. To our knowledge, there exists no prior work that
identifies and estimates causal effects when given a general differentiable functional confounder.

**[R1: Why is positivity violated?]** First note that we use $\mathbf{z} = h(\mathbf{t})$ to denote that the confounder's value $\mathbf{z}$ is provided
by a "part" of the pre-outcome variables $\mathbf{t}$. Positivity is when for any $t \in \mathrm{supp}(\mathbf{t})$, the conditional support of $\mathbf{z}$ given
$\mathbf{t} = t$ exactly matches the marginal support of $\mathbf{z}$. Positivity is violated in EFC because
$$\forall t, t_2 \in \mathrm{supp}(\mathbf{t}) \ s.t. \ h(t_2) \neq h(t) \implies p(\mathbf{z} = h(t_2) \mid \mathbf{t} = t) = 0 \neq p(\mathbf{z} = h(t_2)) > 0$$
In words, two different confounder values cannot occur for the same $t$. We have added this to the paper.

**[R1, R2, R3: Examples in the presentation]** One example of causal redundancy is the assumption underlying existing
GWAS methods like the PCA correction (Price et al.). For the PCA correction, as we discuss in the appendix B.1, the
functional confounder is the genotype projected onto axes of genetic variation: $h(\mathbf{t}) = A\mathbf{t}$. Then, causal redundancy is
satisfied when the outcome function is $f(\mathbf{t}, h(\mathbf{t})) = A_p\mathbf{t} + h(\mathbf{t})$ and $A_p \perp A$. We have added this running example in
the paper.

**[R3 : Examples of surrogate interventions]** Surrogate interventions are elements in the space that $\mathbf{t}$ lies in and not
a distinct type of intervention. Their effects are defined like any other $t \in \mathrm{supp}(\mathbf{t})$. For example, with $\mathbf{t}^i$ as the $i$th
coordinate of $\mathbf{t}$ consider the SCM (as in section 2.1)
$$\mathbf{t} \in \mathbf{R}^2, \quad h(\mathbf{t}) = \mathbf{t}^1 - \mathbf{t}^2, \quad \mathbf{y} = \mathbf{t}^1 + \mathbf{t}^2 + h(\mathbf{t}).$$
Consider we want the true conditional effect for $t = [1, 1]$ and $h(t_2) = 1$:
$$\phi(t = [1, 1], h(t_2) = 1) = t^1 + t^2 + h(t_2) = 3.$$
The surrogate for this $t, h(t_2)$ would be $t' = [1.5, 0.5]$ because its confounder value and conditional effect match $h(t_2)$
and $\phi(t = [1, 1], h(t_2) = 1)$ respectively:
$$h(t') = 1.5 - 0.5 = 1 = h(t_2) \quad \text{and} \quad \phi(t', h(t')) = 3 = \phi(t = [1, 1], h(t_2) = 1).$$

**[R3, Applicability of LODE]** We thank the reviewer for raising an important concern about applicability.

Our work provides a formal basis to reason about the kinds of effects a practitioner can estimate given a general
confounder function. Functional confounders are present in areas other than genetics. For example, consider estimating
the effect of intervening on blood pressure or cholesterol on an outcome like coronary heart disease. Here a confounder
would be Metabolic Syndrome which is defined as a function of cholesterol, blood pressure, blood sugar, and related
measurements. Making this precise with real data would take care and is beyond the scope of the paper. Similar
examples exist for other syndromes that are defined as a collection of variables that can be intervened on individually.

**[R3, LODE adjusts for confounding factors]** We agree with the reviewer that we do not show "correct" adjustment
for confounding: this would require estimated effects as ground truth. Such GWAS ground truth effects are unavailable.

However, the SNPs we report in table 1 in the main paper and table 2 in the appendix have all been reported as relevant
to celiac disease in established GWAS studies. The SNP rs2237236 has a 0 Lasso coefficient meaning that its effect
would be 0 without correction, going against its established relevance. With LODE's correction, we recover this SNP
as relevant. We believe that such correction atop our recovery of relevant SNPs is evidence that LODE corrects for
confounding to an extent sufficient to separate relevant SNPs from irrelevant ones. We will change the language to
make to make this more clear.

[Meta-Review · NeurIPS 2020]

The reviewers all agree that this is a paper worthy of acceptance. The presentation should be improved based on the remarks in the reviews and in particular the discussion around why the positivity assumption is violated in the setting that is studied here (which is a key point) should be made more clear.